# Limitations and Possibilities of Transarterial Chemotherapeutic Treatment of Hepatocellular Carcinoma

**DOI:** 10.3390/ijms222313051

**Published:** 2021-12-02

**Authors:** Charlotte Ebeling Barbier, Femke Heindryckx, Hans Lennernäs

**Affiliations:** 1Department of Surgical Sciences, Section of Radiology, Uppsala University, 752 36 Uppsala, Sweden; Charlotte.Ebeling_Barbier@radiol.uu.se; 2Department of Medical Cell Biology, Uppsala University, 751 23 Uppsala, Sweden; femke.heindryckx@mcb.uu.se; 3Department of Pharmaceutical Biosciences, Uppsala University, 751 23 Uppsala, Sweden

**Keywords:** hepatocellular carcinoma, transarterial chemoembolization, drug delivery systems, tumour microenvironment, anthracyclines

## Abstract

Because diagnostic tools for discriminating between hepatocellular carcinoma (HCC) and advanced cirrhosis are poor, HCC is often detected in a stage where transarterial chemoembolization (TACE) is the best treatment option, even though it provides a poor survival gain. Despite having been used worldwide for several decades, TACE still has many limitations. First, there is a vast heterogeneity in the cellular composition and metabolism of HCCs as well as in the patient population, which renders it difficult to identify patients who would benefit from TACE. Often the delivered drug does not penetrate sufficiently selectively and deeply into the tumour and the drug delivery system is not releasing the drug at an optimal clinical rate. In addition, therapeutic effectiveness is limited by the crosstalk between the tumour cells and components of the cirrhotic tumour microenvironment. To improve this widely used treatment of one of our most common and deadly cancers, we need to better understand the complex interactions between drug delivery, local pharmacology, tumour targeting mechanisms, liver pathophysiology, patient and tumour heterogeneity, and resistance mechanisms. This review provides a novel and important overview of clinical data and discusses the role of the tumour microenvironment and lymphatic system in the cirrhotic liver, its potential response to TACE, and current and possible novel DDSs for locoregional treatment.

## 1. Introduction

Hepatocellular carcinoma (HCC) is a primary liver cancer that usually develops in patients with an underlying liver disease and sustained liver damage. It is one of the most common and deadly cancers worldwide, currently ranking fifth and third in terms of global incidence and mortality of all cancers [1,2]. With the increasing prevalence of obesity, the incidence of liver disease is expected to rise and to include a younger population. In 2020 approximately 906,000 new cases and 830,000 deaths has been reported for primary liver cancer with more than 80% of these from HCC [3], which is expected to become the second leading cause of cancer-related deaths in the US [4,5].

HCC develops after a period of prolonged hepatic damage, leading to inflammation, fibrosis and eventually cirrhosis, the end-stage of chronic liver disease. The occurrence of fatty liver disease is one of its most common risk factors, and it evolves from intrahepatic fat accumulation through steatohepatitis to cirrhosis [6]. Cirrhosis causes increased mortality in various ways; by generating portal hypertension, by progressing into liver failure, and by fuelling the development and progression of HCC. Prevention and treatment can be applied at several stages in this evolution, but disease detection needs to be improved and there is still a vast need for effective treatments. Potentially curative therapies such as liver resection, radiofrequency or microwave ablation, and transplantation are mainly available for patients with early-stage HCC, which has an overall 5-year survival rate reaching 70–80% [7]. Unfortunately, diagnostic tools for discriminating between HCC and advanced cirrhosis are poor [8,9]. Thus, HCC is often detected in the intermediate stage [10], where transarterial chemoembolization (TACE) is the recommended first-line therapy [11,12,13], according to European and American guidelines [5,14,15]. TACE can also be used in early-stage HCC as a bridge to transplantation or when surgery is not possible, and in selected patients with advanced stage HCC [16,17].

TACE is a locoregional chemotherapy performed though a percutaneously inserted intravascular catheter, which has been advanced using image-guidance and positioned in the hepatic artery as close as possible to the liver tumour region to create a local hot spot with the active drug (Figure 1). There are two drug delivery options: conventional TACE (cTACE) and drug-eluting microparticle TACE (DEM-TACE). Key processes related to the success of TACE are drug release and mass transport across the vascular barrier and through the complex and highly dynamic tumour microenvironment.

The rationale for cTACE is that the intra-arterial injection of a viscous emulsion in tumour feeding vessels will result in a strong cytotoxic effect. The emulsion-based product is made from a chemotherapeutic drug (doxorubicin, idarubicin, epirubicin, mitomycin C, or cisplatine) mixed with iodized oil (Lipiodol^®^ Ultra-Fluide, Guerbet, Paris NORD 2, France), and is often followed by embolization of the blood vessel with gelatine sponge or solid embolic agents. This vascular occlusion reduces local blood flow, thereby depriving the tumour from its oxygen and nutrient supply and increasing the local transit time of the formulation that should enhance the tumour’s chemotherapeutic exposure. However, clinical evidence on an enhanced anti-tumour effect caused by the addition of vascular occlusion remains uncertain [18,19]. Contrarily, intra-tumoral hypoxia has been observed to induce angiogenesis, and hypoxic tumour cells are known to become more aggressive and pro-metastatic [20]. A meta-analysis has reported that the expected overall median survival was increased to about 20 months with cTACE compared to 16 months for symptomatic treatment [21].

In DEM-TACE, the chemotherapeutic drug (doxorubicin) is loaded into microspheres from which it is released slowly. However, the microspheres cannot cross the peribiliary capillary plexus and enter the tumour, and thus, the drug is released in the distal arterioles and distributed from there via diffusion, convection, or both [17,22]. The local release of the drug from the microsphere formulation to the vascular site is affected by the microsphere size and loading degree. In vitro experiments have demonstrated that only 30% of the drug dose is released [23,24]. The loading capacity and efficiency also differ between different chemotherapeutic drugs and types of microspheres, causing a significant variability in this locoregional formulation approach. Therefore, more in-depth experimental studies and clinical trials of various combinations of drugs and microspheres are needed to explore the correlation between microsphere size and optimal drug loading and its effects on formulation properties. It has been suggested that half-loaded microspheres with doxorubicin are associated with greater treatment response and shorter hospital stays [25]. The fact that no survival benefit has been detected for patients treated with DEM-TACE compared to embolization with microspheres without the drug [26,27,28] further highlights the need for increasing the understanding of the exact contribution of ischemia to anthracycline-induced cytotoxicity following DEM-TACE [29].

Several randomized control trials (RCTs) and observational studies report no difference in tumour response and overall survival (OS) between cTACE and DEM-TACE, thereby suggesting that both treatments have a similar effectiveness [30,31,32]. However, regarding safety, DEM-TACE entails an increased incidence of hepatobiliary injury in several cohorts, [33,34,35] and less post-procedural pain compared to cTACE in one study [31]. The hepatobiliary injury is probably caused by the coagulative necrosis seen around the microspheres [22], which are wedged in the arterioles at various distances from the tumour, depending on the size of the microspheres. Conventional TACE has been shown to increase OS compared to best supportive care (mean 28.7 months vs. 17.9 months) [36]. Yet the survival increase gained by TACE remains poor (median 19 months) [37] and needs to be improved.

If the delivered drug(s) do not sufficiently penetrate both selectively and deeply into the tumour and the drug delivery system is not releasing the drug(s) at an optimal clinical rate, it may contribute to an overall modest antitumour effect. Furthermore, the effectiveness of TACE is confounded by variations in the HCC genotype and phenotype, and the highly heterogeneous population of patients with intermediate-stage HCC [10]. In addition, the crosstalk between the tumour cells and components of the cirrhotic tumour microenvironment also limits therapeutic effectiveness. Even though TACE has been used for several decades, there has been minor progress in the understanding of the complex interactions between drug delivery, local pharmacology, tumour targeting mechanisms, liver pathophysiology, patient and tumour heterogeneity, and resistance mechanisms [10,37,38]. These factors are all likely to have contributed to the lack of treatment success for therapies targeting HCC [39]. The absence of an overall benefit, together with the generally poor prognosis of patients with intermediate and advanced stage HCC, result in a major medical need.

In this review, we have provided a novel and important overview of clinical data, as well as a deeper understanding of the underlying biological mechanisms that could contribute to the response to TACE. In addition, we also highlight the importance of the tumour microenvironment and lymphatic system in this complex interplay. Therefore, the objective of this review is to discuss three key features: (i) the role of the tumour microenvironment and lymphatic system in the cirrhotic liver and its potential response to TACE; (ii) some of the most frequently clinically applied parenteral products for locoregional drug treatment, and finally; (iii) possible novel drug-delivery systems (DDS).

## 2. The Role of the Tumour Microenvironment

### 2.1. The Elusive Transition from Cirrhosis to Tumour

The microenvironment in a cirrhotic liver stimulates tumour growth and drug resistance, mainly through altered biomechanical properties, secretion of cytokines, and activation of multiple signalling pathways. These properties of the microenvironment contribute to the high rate of recurrence after TACE [40]. TACE is traditionally considered a palliative treatment and it has not been established to have any preventive effects, such as suppressing tumour evolution in the cirrhotic parenchyma. However, the combination of cTACE and radiofrequency ablation (RFA) entails a better survival than RFA alone in an RCT including 189 patients [41]. Moreover, the long-term prognosis (i.e., tumour progression, recurrence, and survival) has been reported to be similar in patients with single 2–3 cm HCCs treated with a combination of cTACE and RFA compared to surgical resection alone [42], despite RFA and surgical resection being considered as curative treatments. This indicates that there may be several minimal HCC foci in the diseased liver parenchyma that are not detectable with currently used imaging or biomarkers, but that receive treatment in addition to what was intended. It is well-known among interventional radiologists (IRs) performing TACE that Lipiodol^®^ accumulation is sometimes seen in such unexpected sites during or after TACE, or both (Figure 2). This highlights the need for better understanding of how the cirrhotic microenvironment actively fuels tumour growth and whether anti-fibrotic treatments could improve the outcome in HCC patients. These trials also support that the standard of care for HCC suitable for RFA should be a combination treatment with locoregional chemotherapeutics.

The tumour microenvironment is known to be an active contributor to the progression of HCC. It is therefore unsurprising that several treatments have been developed that target the dynamic fibrotic environment that surrounds HCC-nodules. For instance, drugs targeting TGF-β (LY2157299), which is a growth factor highly involved in stellate cell activation, have been tested in clinical trials (NCT01246986 and NCT02178358). Another important contributor to the microenvironment is the deposition of ECM, which includes heparan sulfate proteoglycans. PI-88 is a heparin sulfate mimic that specifically targets heparanase in cancer, thus preventing the release of growth factors that otherwise would contribute to tumour growth, angiogenesis, and metastasis. The safety and efficiency of PI-88 as an adjuvant therapy for post-operative HCC has been shown in a phase II trial, and a recent follow up study revealed significant clinical benefits for patients with HCC (PMID: 1930316), which has been further studied in a phase III trial (NCT01402908). However, none of these compounds have been tested in combination with TACE, yet it could be speculated that reducing the dense network of activated stellate cells and deposition of ECM, would positively influence the response to TACE in patients. More studies are needed to confirm this.

Significant progress on the treatment of advanced HCC has been made possible by sorafenib, which in itself targets the tumour microenvironment by blocking angiogenesis. While TACE and sorafenib have both been shown to prolong survival in patients with unresectable hepatocellular carcinoma, previous trials assessing the combination of TACE and sorafenib have been unable to show clinical benefits compared with TACE alone. A recent study from Kudo et al. reports the results from a randomized, multi-centre prospective trial of this combination treatment in patients with unresectable HCC. Median progression-free survival was significantly longer in the combination group (25.2 months) than in the TACE without sorafenib group (13.5 months), thus showing promising results in tackling both the tumour cells and the microenvironment in the context of TACE.

### 2.2. Cellular Interactions and Immunology

The tumour microenvironment contains non-malignant cells that are assisting tumour cells to proliferate, invade, and metastasize. Furthermore, immunosuppression, associated with chronic inflammatory factors, such as growth factors, cytokines, and chemokines is generated by both stromal and tumour cells [43,44]. Multiple immune cells coexist and interact in a parallel and advanced sequence of signalling pathways, thereby fuelling carcinogenesis.

The cirrhotic liver is characterized by a severe inflammatory reaction and an excessive deposition of extracellular matrix, which actively contributes to the progression of HCC [45,46]. Several studies have suggested that therapeutic response relies partly on extrinsic mechanisms provided by the crosstalk between tumour cells and components of the tumour microenvironment (Figure 3). The tumour microenvironment may play a role in the initiation and maintenance of drug resistance through various mechanisms, such as alterations in the pH, presence of hypoxia, abnormal vasculature, changes in the immune populations, and the activation of hepatic stellate cells, including their secretomes, the extracellular matrix, and other soluble factors [47].

The hepatic immune population is highly heterogeneous. It is comprised of myeloid-derived suppressor cells, regulatory T cells, and tumour-associated macrophages, which are responsible for the building of an immunosuppressive pro-tumoral environment. The intrinsic immunosuppressive nature of the hepatic microenvironment plays a major limiting role in the effectiveness of chemotherapeutic agents. In addition, chemotherapeutic agents themselves alter the composition of the hepatic inflammatory populations.

The most dominant immune cells of the tumour microenvironment are tumour-associated macrophages, which have been established as key mediators of drug resistance in several solid tumour types [45,48]. It has been shown that the density of tumour-associated macrophages in biopsies is associated with the efficacy of TACE in HCC patients, possibly by inducing autophagy in the HCC-cells, which could inhibit Oxaliplatin-induced cell death [49]. TACE itself can significantly modulate the immune response by inducing immunogenic cell death, which is a form of cell death resulting in a regulated activation of the immune response.

Studies have shown that immunogenic cell death involves the release of calreticulin and other endoplasmic reticulum proteins, as well as the secretion of ATP, and apoptosis-associated release of the non-histone chromatin protein high-mobility group box 1 [50]. This stimulates recruitment and activation of dendritic cells near the tumour site, which will allow engulfment of tumour antigens from dying tumour cells and optimal antigen presentation to T-cells [51]. Doxorubicin, the most commonly used TACE drug, has been shown to induce apoptosis directly leading to immunogenic cell death and immune activation in the liver [52]. More specifically, an increased CD4/CD8 ratio, an increased number of th17 cells, and a marked decrease in T-regs characterize the peripheric immune population after TACE treatment, supporting a favourable immune profile [49]. In addition, there is an increased release of pro-inflammatory cytokines, such as IL-22 and IL-6, early after TACE treatment; and in patients with larger tumours (>5 cm) there is an early increase in IL-6 and a late increase in IL-4, IL-5, and IL-10 after TACE treatment [53].

The well-established contribution of the immune-cell population in the pathogenesis of HCC has led to several clinical trials assessing the potential of immunotherapy in HCC patients [54,55]. Studies investigating the expression of immune checkpoint regulators after TACE present conflicting evidence on the role of programmed death receptor 1 (PD1) or programmed death ligand 1 (PD-L1). The identification of PD1 as an important prognostic marker [56] is contradicted by a study demonstrating overall low levels of PD1 after TACE and instead increased levels of another inhibitory immune checkpoint, i.e., T-cell immunoglobulin and mucin domain-3 (TIM-3), with higher levels in complete responders than in partial responders [55]. The expression of TIM-3 on various innate and adapted immune cells has suggested its involvement in a rather complex range of immune pathways that interact with the PD1/PD-L1 axis. Thus, TIM-3 may have a role in anti-tumour immunity after TACE, possibly though interaction with the PD-1 pathways, suggesting a possibility to combine immunotherapy and chemoembolization. [55]

Overall, these observed alterations in inflammatory populations and inhibitory immune checkpoints after TACE treatment warrant further research on whether a combination of TACE with chemotherapeutics and immune-checkpoint inhibitors could increase drug-response.

### 2.3. Vascularity and Penetrability

In addition to the cellular interactions, the cirrhotic microenvironment forms a physical barrier that prevents the drugs to reach their target(s) within the tumour parenchyma, which reduces drug availability [47]. The tumour microenvironment is characterized by an abnormal vascular network [57], an increased interstitial fluid pressure, excess extracellular matrix (ECM), and abundant tumour-associated stromal cells that all, to various extents, affect drug transport and limit drug availability [58]. Following an intravascular administration, the tortuous tumour vasculature, uneven blood flow distribution, heterogeneous leakiness, and high interstitial fluid pressure present a primary barrier for any drug formulation from the vascular compartment to the tumour site [59].

However, vascular leakiness can increase the perfusion of chemotherapeutic drugs at the local tumour site. A key process for a successful tumour targeting formulation is an enhanced permeability and retention (EPR) effect. This effect has been extensively investigated as a unique phenomenon of solid tumours related to their anatomical and pathophysiological differences from normal tissues, and it is fundamental for the development of macromolecular and nanoparticle based anticancer therapy. Unfortunately, progress in developing effective pharmaceutical formulations, especially particles in the nano-size range, has been hampered by a heterogeneity in the EPR effect and a lack of information on factors that influence this process in the liver [60,61].

The tortuous and disorganized nature of the tumour vasculature complicates an even drug distribution. In addition, HCC is severely affected by the hypoxia induced by the intra-arterial formulation applied in TACE, which may induce angiogenesis and/or alter the complex cellular metabolism and thereby reduce the treatment effect. This mechanism of increased resistance to treatment might be reduced by combining chemotherapeutic drugs with anti-angiogenic and anti-proliferation treatments, such as the multiple kinase inhibitor sorafenib [62].

The abundant ECM and the substantial stromal cells, such as activated hepatic stellate cells and tumour-associated macrophages, are considered a binding-site barrier that may reduce the penetration of drug formulations within solid tumours [47]. This leads to an accumulation of drugs around the blood vessels, which is unintendedly internalized by stromal cells. This excessive deposition of the ECM also alters the biomechanical properties of the diseased liver, leading to increased liver stiffness, which in itself enhances tumour growth, decreases drug response, and limits drug diffusion and availability [46].

Drug mass transport and availability at the core of a solid tumour is significantly influenced by the elevated interstitial fluid pressure and poor vascularization [63]. Both cTACE and DEM-TACE are considered to produce a better anti-tumour response in tumours with a high degree of vascularity [64]. A correlation between a longer doxorubicin exposure time and lower cytostatic effect (IC50) has been established in four human cancer cell lines (HepG2, Huh7, SNU449 and MCF7) [65]. This can be explained by an increased cellular uptake and intracellular exposure of doxorubicin, which was observed to accumulate intracellularly where concentrations were 230 times higher than exposure concentrations [65]. This high intracellular accumulation of doxorubicin and the lack of a distinct correlation with the extracellular concentration suggests that passive diffusion is a key mechanism for the overall intracellular delivery of doxorubicin into tumour cell lines [65]. The high intracellular concentration of doxorubicin can be explained by the high binding and retention capacity of the drug to nuclear and mitochondrial DNA, which has been proved to be its main intracellular binding-sites in both experimental and quantitative pharmacological modelling [66,67]. However, after a TACE treatment this high, intracellular accumulation of doxorubicin entails a decreased intercellular diffusion, which may reduce the penetration of the drug and its effect in avascular tumour regions.

Thus, diffusion remains the main driving force for drug transport in these heterogenous tumours, and the diffusion process needs to be better understood to develop new drug molecules, nano particles, and other advanced DDS.

### 2.4. The Potential Role of Liver Lymphatics

The liver is the largest lymph-producing organ in the human body, generating about 50% of the body’s total lymphatic fluid, and it is the most important part of the lymphatic system from a functional point of view [68]. The main circulatory role of the lymphatic system is to transport fluid back from the tissue into the venous system. Therefore, understanding the anatomy, physiology, and pathophysiology of the lymphatics of the liver is important to better understand how it affects the tumour uptake, the distribution, and the elimination of various chemotherapeutic drug formulations. The lymphatic system has been poorly studied and, unfortunately, it is not easily depicted with standard imaging techniques.

It has been reported that the volume of the lymphatic fluid produced in a cirrhotic liver is up to 30 times larger than in a non-cirrhotic liver [69]. Resistance to sinusoidal blood flow increases in cirrhotic livers because of morphological changes around the portal and central venules. The sinusoidal hydrostatic composition leaks fluid from the blood vasculature into the interstitial space through highly permeable liver sinusoidal endothelial cells and this forms the lymphatic fluid. It has also been reported that lymphangiogenesis occurs in liver diseases as a response to vascular endothelial growth factors (VEGF) expressed by infiltrating macrophages [70]. During end-stage liver disease, lymph production and lymphatic vessel frequency increases in both human livers, as in animal models [71]. This increased frequency of lymphatic vessels occurs alongside an increased expression of VEGFC/D in the cirrhotic liver [72]. Furthermore, patients with cirrhosis have been reported to have an increased volume of lymph fluid draining the liver [73]. It seems plausible that the lymphatic filtration would be increased in a cirrhotic liver, which would increase the clearance of the chemotherapeutic drug and consequently reduce its transit time and availability at the liver tumour site. However, it is unlikely that there would be functional lymphatic vessels inside a tumour. Thus, the effects of the increased lymphatic vascular flow in the liver and its effect(s) on the effectiveness of locoregional drug therapies is an area for future investigation in this patient group.

## 3. Current Drug Performance

Apart from bland embolization, aiming to starve the tumour by inducing ischemia with particles, or degradable starch microspheres (DSM), there are two main types of parenteral drug delivery systems (DDS) for TACE, which provide a transient or permanent vascular occlusion-mediated ischemic reaction, in combination with drug-mediated cytotoxicity: cTACE and DEM-TACE.

### 3.1. Conventional TACE (cTACE) with Chemotherapeutics in Lipiodol^®^ Emulsions

The viscous emulsion used for cTACE is composed of Lipiodol^®^ (Guerbet, France) and a wide variety of chemotherapeutic drugs. Both single-drug and fixed-dose drug combinations are used, where doxorubicin is the most common [74]. The Lipiodol^®^-based emulsions can be administered in lobular, segmental, or subsegmental arteries, and all administration sites are used in a clinical context. Selective or super-selective drug administration has been recommended to optimize tumour necrosis and reduce adverse effects to the non-tumour liver parenchyma [75].

Lipiodol^®^ is an iodinated poppy seed oil, with 73% line-olic, 14% oleic, 9% palmitic, and 3% stearic acid, which makes it suitable as a contrast agent. It was clinically introduced in radiology in the 1920s as an X-ray marker, and in the 1980s it was reported to accumulate in rabbit tumour tissue [76]. When the emulsion is administered in the hepatic artery of rats, Lipiodol^®^ appears in the portal venules, and passes through to the sinusoids. This causes a temporary and partial stasis of blood flow [77]. These tumour–seeking and embolization properties have formed the rationale for its clinical use in cTACE treatment of HCC [76]. After intrahepatic administration of the Lipiodol^®^-based drug formulation, it is also feasible to administer embolic particles that can be biodegradable (e.g., gelatine sponge) or permanent (e.g., polyvinyl alcohol microparticles) to further enhance the ischemic effect [78].

Besides being the vehicle for drug delivery, Lipiodol^®^ may also function as an imaging biomarker for tumour response because the necrosis has been proved proportional to the fraction of tumour volume opacified by Lipiodol^®^ on computer tomography (CT) [79]. The proposed mechanisms for lipiodol accumulation in tumour cells are: (i) cell membrane pumps; (ii) pinocytosis, (iii) reduced degradation by lysosomes [78,80], or a combination thereof.

Lipiodol^®^ can also be used as an imaging biomarker for survival, because clinical trials have demonstrated improved survival in patients with more complete Lipiodol^®^ retention in the tumour [81,82]. Furthermore, patients in whom peritumoral portal Lipiodol^®^ enhancement (PPLE) is seen during cTACE have longer survival and better tumour response than patients without PPLE [83]. An intense PPLE is also associated with a greater extent of histopathological necrosis [84] and a lower local recurrence rate [85,86]. Presumably, PPLE reflects the extent to which the tumour has been permeated with the emulsion and, thus, exposed to the drug. However, it cannot be predicted in which patients or to what extent PPLE will occur, and no technical trick has been identified as how to achieve it. The occurrence of PPLE might rather be governed by morphological factors of the tumour and its microenvironment. For instance, it has been suggested that Lipiodol would more easily pass through a moderately than a poorly differentiated tumour and thereby provide a greater ischemic effect, because there is a more complete outflow tract [87]. However, no correlation between tumour differentiation and the effect of TACE has been reported. Furthermore, cell transformation can occur at any stage from progenitor to adult hepatocytes, giving rise to HCC of varying morphologies [88]. Progression of liver disease and the development of HCC are associated with an overexpression of β-catenin [89], which has been established as a prognostic factor for survival in HCC patients treated with TACE [90].

Visible uptake and retention during the cTACE procedure varies greatly between studies [91]. A variety of different lipiodol^®^-based emulsions have been used clinically, and the full composition of the formulation is often not described in reports [92], thereby complicating the interpretation of results obtained from different studies. In addition, every step in composing Lipiodol^®^-based emulsions constitutes a source of variation. For instance, the final emulsion ex tempore preparation is often performed in the IR suite at ambient temperatures. The emulsion is then prepared by using a simple pumping technique, which has been identified as a critical formulation step, affecting the properties and performance of the final DDS [23,78,92]. However, it is unlikely that this step is performed in exactly the same manner by all IRs, thereby increasing variability. Because the composition of the emulsion is not standardized, there are large variations in preparation technique, aqueous-to-lipid phase ratio, aqueous phase composition, and injectability properties [78,93]. All these parameters affect the drug release rates [93] and can further introduce variability between different studies.

For example, it has been suggested that the addition of a contrast agent to the aqueous phase increases emulsion stability and decreases drug release rates [94]. The proposed mechanism is that the contrast agent generates similar densities to the emulsion phases, which reduces the separation rate. Furthermore, adjustment to the aqueous-to-lipid phase ratio by increasing the Lipiodol^®^ part (1:2–1:4) enhances the emulsion stability and reduces the drug release rate from the formulation both in vitro and in vivo [92,95]. A recent study reported that the emulsion stability and in vitro release profiles of doxorubicin are both affected by the composition of the aqueous phase and the aqueous-to-lipid phase ratio [96]. In addition, the relatively low distribution and solubility of doxorubicin to the lipid phase suggest a need for improvements of this DDS.

These variations, along with the lack of information on emulsion composition in clinical reports, makes comparisons of tumour response between clinical studies highly difficult. This probably contributes to the fact that there is still no consensus regarding the optimal chemotherapeutic drug for TACE. Despite more than six decades of experience in the clinical use of anthracyclines, their mechanism(s) remain mainly empirical [93].

The dose of the active drug and its concentration-time profile in local liver tissue, tumour tissue, and plasma has not been well characterized for either of the drugs used in the treatment of HCC. It is clear that the fraction of the Lipiodol^®^ dose that is retained in the tumour is related to the tumour size, tumour physiology, and biochemical condition of different liver compartments. However, it is unclear which dose of the active drug in these formulations provides the best benefit-to-safety ratio. Besides doxorubicin, the most used chemotherapeutic drugs for cTACE are epirubicin, cisplatin, miriplatin, and idarubicin.

As idarubicin is more lipophilic and has a smaller polar molecular surface area than doxorubicin, a higher idarubicin concentration can be achieved in Lipiodol^®^, entailing a higher effective permeability across biological membranes [97]. The phase separation for a Lipiodol^®^ - idarubicin based emulsion has been reported to be very limited (5% aqueous solution and 95% persisting emulsion); fluorescent electronic microscopy has revealed idarubicin to be located inside and at the interface of the emulsion because of drug-lipid ionic interactions. This enhanced emulsion stability is expected to be beneficial as it increases the contact time of the drug with the cancer cells [98]. Based on in vivo data in an advanced multiple-sampling pig model, the local liver distribution of doxorubicin is affected more by formulation-related properties of the emulsion than by any direct interactions between Lipiodol^®^ and membrane barriers or transport proteins [99]. Doxorubicin is released very rapidly from the emulsion both in pigs and in human HCC patients [38,99]. This is most likely explained by the poor stability of the doxorubicin–Lipiodol^®^ emulsion. Because Lipiodol^®^ is known to be retained by HCC cells for at least several weeks, a more stable formulation could provide a slow-releasing, tumour-targeting vector. Thus, idarubicin, which forms a more stable emulsion with Lipiodol^®^, could be a suitable alternative.

Furthermore, in vitro studies on three HCC cell lines have demonstrated idarubicin to be the most effective among 11 tested chemotherapeutic drugs, including doxorubicin, cisplatin, and epirubicin [100]. The superiority of idarubicin in terms of cytotoxicity was most notable in chemo-resistant HCC cells, which strongly suggests that idarubicin has a therapeutic benefit over doxorubicin. Clinical studies have reported similar efficacy and safety for idarubicin and doxorubicin-TACE [101,102]. Thus, idarubicin-TACE may represent an interesting alternative to doxorubicin-TACE in the management of patients with intermediate stage HCC.

### 3.2. Lipid Nano-Formulations with Anthracyclines

Liposome based nano-formulations have been widely investigated as carriers for potent chemotherapeutic drugs with the objective to increase efficacy and reduce toxicity. An example is the encapsulation of doxorubicin within PEGylated liposomes. The intra-particle drug load of doxorubicin is more than 90% in the nano-liposome, which has a size range of 80–100 nm in diameter. The proposed non-specific target mechanism for nano-sized formulations in solid tumours is mediated by the EPR effect.

While nano-formulations have been reported to exhibit improved anti-tumour effects in a variety of animal models, the translational and clinical relevance of the EPR effect of nano-formulations has been challenged. In patients, treatment with doxorubicin in nano-sized PEGylated liposomes produced a prolonged total plasma exposure and improved safety properties but did not improve efficacy compared to systemic intravenous treatment with doxorubicin. In patients with advanced HCC, treatment with PEGylated liposomal doxorubicin after intravenous administration of gemcitabine has been demonstrated to be active and safe [103]. Thus, single therapy with PEGylated liposomal doxorubicin has not yet been proven to be clinically effective in treating any stage of HCC [103].

It is well-established that liposomal nano-formulations with doxorubicin have a strong effect on the total plasma and blood concentration-time profile and tissue distribution. The tissue distribution of encapsulated and free doxorubicin reflects the direct deposition of doxorubicin-based nano-liposomes. Tumour vascular endothelium and ECM are considered as major barriers for tissue distribution of therapeutic nanoparticles and can, consequently, limit the anticancer effect. There are few strategies available to overcome this rate-limiting step. Doxil^®^ is a liposomal nano-formulation, which penetrates much less into ECM, resulting in a lower exposure of doxorubicin intracellularly, which reduces the frequency of its most severe adverse effects in the heart and intestine. The maintained anti-tumour effect and lower frequency of these side-effects are consequences of a longer terminal half-life due to the poor vascular penetration and encapsulation of doxorubicin, which is the most significant clinical gain of this PEGylated liposomal product (Doxil^®^). There are experimental strategies that utilize a reversible vasodilatation effect mediated by nitric oxide and an ultrasound responsive liposome able to cross both rate-limiting steps at the same time [104].

### 3.3. Hydrogel Microparticles (Microspheres) as Drug-Eluting Entities

Hydrogel microparticles are used as a DDS for chemotherapeutic drugs, proteins, and peptides, as they protect these drugs from degradation and also protect the environment from the drug itself, thus decreasing side-effects [105,106].

Several types of DDSs can be used for DEM-TACE, where DCBead™, HepaSphere™, LifePearl^®^, Tandem™, DCB, and HepaSphere™ are approved globally, and there are reports on experimental and clinical investigations of more DEMs [23,107]. DEMs are classified as a medical device, and available in three size ranges between 70 and 700 μm. They are loaded with doxorubicin prior to the TACE procedure and provide a controlled drug release aiming to reduce the systemic anthracycline exposure and subsequently its side-effects.

While doxorubicin remains the most common used anthracycline in DEM-TACE, worldwide doxorubicin shortages challenge the current treatment regime for intermediate stage HCC patients, and finding new alternatives to doxorubicin is essential. DEMs loaded with idarubicin have been demonstrated to have a good safety profile, a promising objective response rate (ORR) and time to progression (TTP) in retrospective [108], as well as prospective cohorts [109]. Compared with prior prospective trials of doxorubicin-based TACE, the ORR and TTP indicate a potential benefit for TACE with idarubicin [97,108]; yet more multi-centre trials are needed to confirm this.

As the DEMs are non-biodegradable, they generate a permanent embolization in the treated arteries, which is considered to contribute to the anti-tumour effect, but may also further worsen the liver toxicity [110]. If the intended dose does not cause full embolization, unloaded microparticles might be used to prevent the blood flow to reach the tumour. The permanent embolization caused by this DDS makes it impossible to treat the same vessel several times, which is possible when using a Lipiodol^®^-based emulsion. Furthermore, ischemia is known to stimulate angiogenesis and might even fuel tumour growth [111].

Compared to the variability of the Lipiodol^®^-based emulsions, the DEMs have been described to generate reproducible and predictable results in blood vessel blockage, and a prolonged drug release. However, the DEMs have not been proved to release the entire drug dose nor to enter the tumour, but they seem to release only parts of the drug [23,24] in arterioles at various distances from the tumour depending on the particle size [17,22]. Thus, it is unclear how much the drug adds to the anti-tumour effect or whether that is mainly caused by ischemia. This uncertainty is supported by the observation that DEM-TACE gives no survival benefit compared to embolization with microspheres without the drug [26,27,28].

Thus, this DDS needs further investigation and development. For instance, the size of the DEMs and the loading dose of the chemotherapeutic drug are expected to affect the treatment response, survival outcomes, and side-effect panorama. Several clinical trials have reported that smaller DEMs (100–300 μm) are associated with a better treatment response and survival, and a reduced number of adverse events than larger DEMs (300–700 μm) [112,113]. A plausible explanation for the improved anti-tumour effects for smaller DEMs might be that they penetrate more readily into the artery and have a higher spatial density, allowing a higher fraction of the dose to be delivered into the deeper tumour and non-tumour tissues [114,115]. However, it is crucial to recognize that other studies report no significant difference in clinical outcomes between beads of size 70–150 μm and 100–300 μm [116].

## 4. The Future of TACE

### 4.1. Preclinical Studies

Most of our understanding of carcinogenesis and drug response in HCC has been derived from mouse models [45,117]. Murine models for HCC have clear advantages in terms of high throughput, as they allow for the treatment of a large number of animals with easy handling, housing, and low costs. However, their small size is a disadvantage when investigating locoregional therapies, due to the disparities in scale and the technical difficulties of performing TACE.

Currently, three animal models are used to investigate the effects of interventional oncology tools. Firstly, the rabbit VX2 model, which has been widely used for intra-arterial drug delivery to hepatic tumours. [118] This model has a few major limitations, as the VX2 cell line is a leporine anaplastic squamous cell carcinoma induced by papilloma virus, which does not resemble most human HCC cell types [119]. In addition, the VX2 model is an orthotopic transplantation model, whereby tumour cells are propagated in the muscle of one donor animal and subsequently transplanted to the liver of the recipient animal. [118] This imposes technical difficulties, in addition to the obvious limitations of not taking in account the crucial influence of the cirrhotic tumour microenvironment on early carcinogenesis, tumour progression and drug response.

In addition, two other commonly used animal models are the rat, where HCC is induced by injections with diethylnitrosamine [120], and the woodchuck, where it is induced by injections with hepatitis virus [121]. In both these models, tumours occur spontaneously in the liver and develop in a background of chronic liver disease, thereby incorporating the important contribution of inflammatory cells, stromal cells, and a fibrotic microenvironment. Further characterization of these models is essential to achieve mechanistic understanding, to develop new formulations for TACE, and to assess new combinational therapies.

To overcome the limitations of current TACE formulations, an aqueous phase of ultrasound-triggered, doxorubicin-loaded, nanoparticle-conjugated microbubble complexes has been emulsified in Lipiodol^®^ and tested on the rabbit VX2 model [122]. This seemed to be more effective than the cTACE formulation, and effectively targeted cancer cells in the tumour periphery.

As described above, the EPR effect is one of the tumour specific pathophysiological processes that is fundamental for the development of macromolecular and nanoparticles anticancer therapies. Yet the heterogeneity in the EPR effect and lack of mechanistic understanding of this process have hampered the development of effective pharmaceutical formulations that would benefit from this tumour-specific condition [60,61]. An alternative to using the EPR effect might be to use local transarterial administration and a possible stimuli-responsive nanocarrier. The efficacy of such nanocarriers depends on the delivery and residence at the intended target site, as well as the degree of vascularity and arterial anatomy. Promising early experimental in vitro results have been reported from a novel asialoglycoprotein receptor targeted PEGylated paclitaxel nanoliposome for HCC that exhibited enhanced cytotoxicity compared to free paclitaxel and improved cellular uptake level based on the EPR effect [123]. Significant research and development is needed to improve the success of these translational findings from in vitro to in vivo and eventually to clinical use. Further research is also warranted to truly understand how the EPR-effect can be utilized to enhance drug delivery at the tumour site.

Despite more than six decades of experience from clinical use of anthracyclines, their mechanism is still mainly empirical and there is no consensus regarding the optimal chemotherapeutic drug of TACE in the treatment of HCC [93]. It is important to note that there is considerable heterogeneity in the in vitro cytostatic effect (IC50) of doxorubicin between different HCC cell lines [29]. This variability in in vitro cell sensitivity to anticancer agents may to some extent explain why partial response is reported in up to 62% of patients treated with TACE [124]. The mechanism for these differences in tumour response remains largely unclear and a mechanistic understanding could enable the discovery of new targeted therapies.

For instance, it has been demonstrated that both idarubicin and doxorubicin increase the levels of polyunsaturated fatty acids (PUFAs) and alkylacylglycerophosphoethanolamines (etherPEs) in different HCC-cell lines with markedly different drug-responses [125]. The fact that these lipids are fundamental in lipid peroxidation during ferroptotic cell death suggests that supplementation of these lipids may have the potential to act as a general adjuvant of idarubicin and doxorubicin in TACE treatment. Ferroptosis is a regulated necrosis that is more immunogenic than apoptosis. This supports a potential synergy between systemic immunotherapy and anthracycline-based TACE, thereby supporting the idea of combining immunotherapy with TACE. As tumours are mainly characterized by highly variable metabolic reprogramming and there is extensive inter-tumour variability of metabolic pathways to support the anabolic demand for tumour growth and proliferation. Thus, an improved understanding of metabolic rewiring will be crucial for the innovation and development of novel drug targets and products.

Modulating the different lipid metabolic pathways could also form a new treatment strategy and contribute to generalized and personalized metabo-chemotherapies. In addition, it has been demonstrated that doxorubicin in itself induces the unfolded protein response pathways in different tumour cell lines [125]. Activation of unfolded protein response is known to induce drug resistance in HCC [126,127]. Thus, combining doxorubicin with small molecules that inhibit the unfolded protein response or endoplasmic reticulum stress pathways could improve tumour cell sensitivity to chemotherapeutics and increase drug response. Although these small molecular inhibitors have shown a good safety profile in preclinical models [45,128], more research is needed to confirm their potential use in patients, especially in combination with TACE.

### 4.2. Clinical Studies

Several attempts have been made to combine TACE with systemic agents, mainly those using multi-kinase inhibitors targeting angiogenic pathways. The SPACE and TACE 2 trials compared TACE combined with Sorafenib to TACE alone [129,130] and similar studies have been performed on Brivanib [131] and Orantinib [132]. Despite the plausible rationale for performing a combinational strategy, all four trials failed to show any clinical benefit of the combination treatments compared to TACE alone. [133] A recent advancement for intermediate-stage HCC with high tumour burden is initial Lenvatinib therapy with subsequent selective TACE, which improves overall survival in an otherwise TACE-unsuitable subpopulation of patients [134]. Thus, Lenvatinib and TACE in sequential therapy may become a valuable option for patients not expected to benefit from TACE alone [135].

In April 2021, a four-year-long clinical trial has started that directly compares Atezolizumab and Bevacizumab to TACE in patients with intermediate-stage HCC [136]. The primary objective is to assess the efficacy (failure of treatment strategy, i.e., disease progression or death) and both cTACE and DEM-TACE approaches are accepted. This is the first head-to-head trial in patients with intermediate stage HCC, but until this study is finalized, TACE is expected to remain the primary treatment for this group of patients.

It is also considered that in the future the combination of TACE and immunotherapy might be very useful clinically. The two main immune checkpoints are the PD-1/PD-L1 and the cytotoxic T lymphocyte-associated protein 4 (CTLA-4). Based on two phase II clinical trials, CheckMate040 and Keynote-224, in which the PD1-inhibitors nivolumab and pembrolizumab yielded promising results as second line agents after sorafenib-treatment, these drugs have become approved as second-line therapies for HCC [54]. However, their potential in combination with TACE remains largely unknown.

## 5. Expert Opinion (TBA)

Despite its shortcomings, TACE has been the standard of care for intermediate stage HCC for several decades [11,12,13]. Numerous factors have contributed to the lack of treatment success [39]. A major limiting factor is the often high and spread tumour burden caused by the fact that early detection of HCC is hardly possible in a cirrhotic liver, despite the constant improvement of imaging techniques. Another major limiting factor is the vast heterogeneity in the cellular composition of HCCs, as well as in the patient population [10], which renders it difficult to identify patients who would benefit from TACE. Furthermore, important knowledge is lacking regarding the complex interactions between drug delivery, local pharmacology, tumour targeting mechanisms, liver pathophysiology, and resistance mechanisms [10,37,38]. There are a number of on-going clinical trials considering various factors and processes that are expected to lead to improvements for targeted and personalized TACE.

Lipiodol^®^ presents several advantages as a DDS in TACE treatment: in addition to being a vehicle for the chemotherapeutic drug, it also acts as an embolic agent, and as a marker for tumour response. The fact that there is no consensus regarding the optimal chemotherapeutic drug for TACE is probably of less importance than the unreliable properties of the emulsions. The great variations in preparation technique, aqueous-to-lipid phase ratio, aqueous phase composition, and injectability properties of the emulsions [78,93] all affect the drug release rates [93]. Furthermore, the variations in choice of drug, dose, number of treatments, emulsion composition, and the lack of information on the latter in clinical reports largely invalidates comparisons of tumour response between clinical trials.

During the last decade, realization of doxorubicin-TACE in patients with intermediate HCC has become complex due to discontinuation of production of lyophilized doxorubicin by many pharmaceutical groups. The use of the lyophilized form is essential to obtain a mix with sufficient viscosity to induce the embolizing effect in cTACE. Currently, only Pfizer continues to produce lyophilized doxorubicin, which has led to recurring shortage periods and a vast need to find treatment alternatives. Idarubicin has exhibited several pharmaceutical formulation and pharmacological advantages and future clinical trials will investigate the clinical response and outcome.

Ideally, the active drug(s) in cTACE or DEM-TACE should, after release in the vicinity of the tumour, have an effective target mass transport into the effect compartment. In addition, an optimal drug for TACE should have a high total body clearance and a large component of clearance outside the target compartment in the liver, to reduce systemic exposure and risk for serious adverse effects. To develop and improve DDSs for TACE, the diffusion process needs to be better understood because it is the main driving force for drug transport. In addition, the effects of an increased lymphatic vascular flow on the effectiveness of locoregional drug therapies is an area for future investigation.

The unsatisfactory effect of TACE can be improved by combining TACE with other locoregional or systemic therapies, which is sometimes performed in the clinical setting. It remains to be seen if a future TACE strategy will include several active components enhancing the anti-tumour effect in different ways. The chemotherapeutic effect might be augmented by drugs modifying the inflammatory and cirrhotic microenvironment, such as immune-checkpoint inhibitors, anti-fibrotic or anti-angiogenic drugs, or by inhibitors of the unfolded protein response or endoplasmatic reticulum stress pathways.

Although there is a constant development and improvement of imaging techniques, early detection of HCC is still challenging. Thus, TACE remains an important treatment tool despite its limitations. The absence of an overall benefit, together with the generally poor prognosis of this common cancer, highlight the need for further research and development, to increase our understanding of how the different tumour and stromal factors contribute to drug response, and to optimize the treatment.

## Figures and Tables

**Figure 1 ijms-22-13051-f001:**
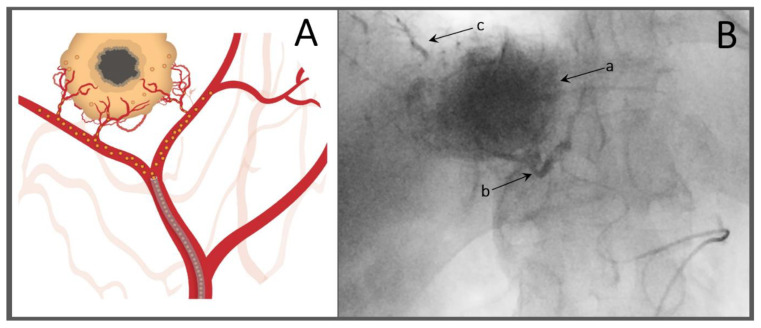
Schematic (**A**) and angiography (**B**) images during transarterial chemoembolization (TACE) through a microcatheter positioned selectively in a tumour-feeding branch of the hepatic artery. The angiography image demonstrates lipiodol accumulation in the hepatocellular cancer tumour (a), stagnant blood flow in the feeding arterial branch (b), and peritumoral portal lipiodol enhancement (c).

**Figure 2 ijms-22-13051-f002:**
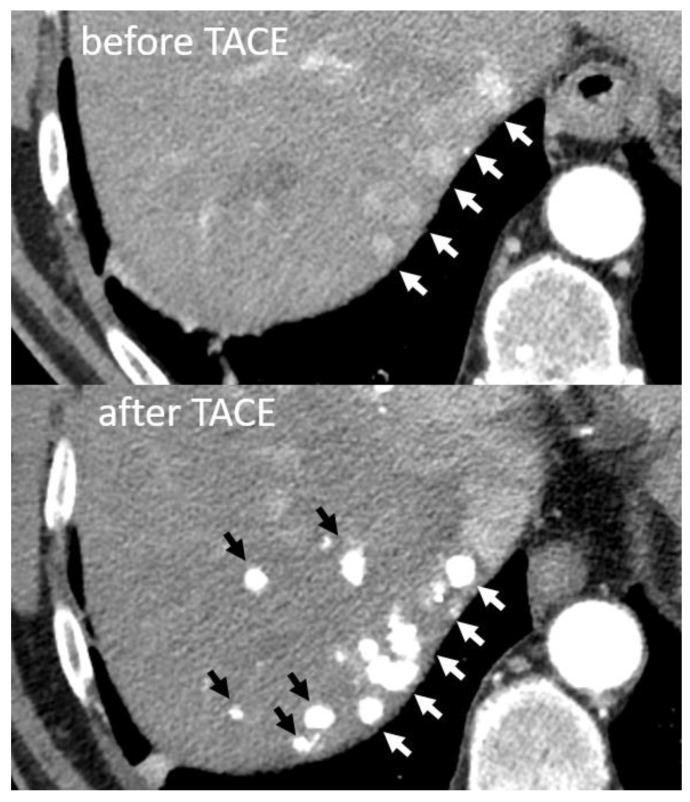
Contrast enhanced arterial phase computer tomography (CT) images displaying lipiodol accumulation in more hepatocellular carcinoma lesions after lobar transarterial chemoembolization (TACE) (black arrows) than what could be detected before TACE (white arrows).

**Figure 3 ijms-22-13051-f003:**
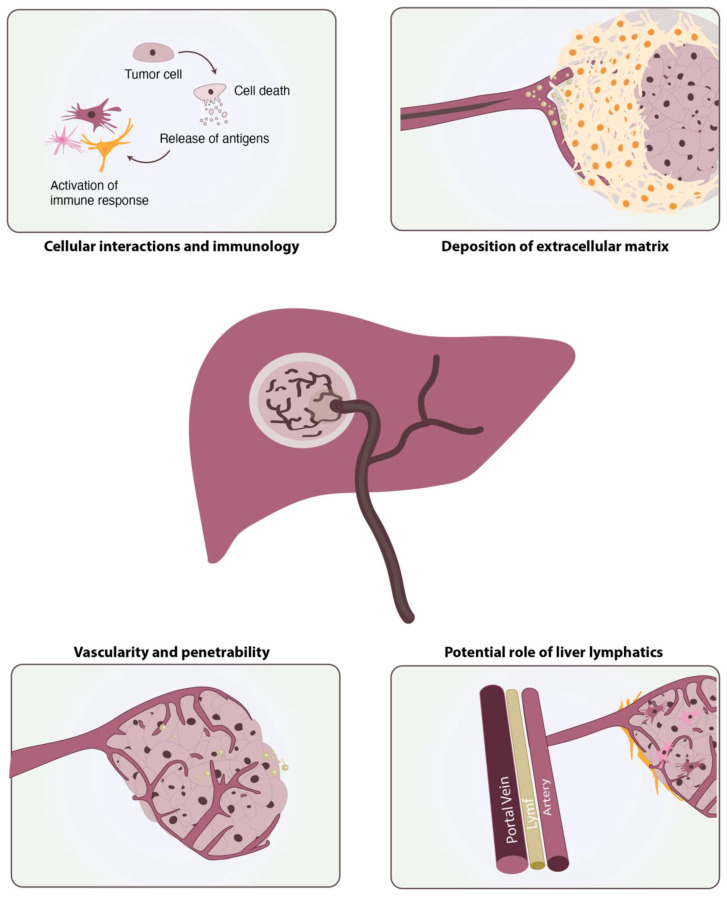
Schematic image demonstrating the crosstalk between tumour cells and components of the tumour microenvironment.

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
