# Peer review of "Limitations and Possibilities of Transarterial Chemotherapeutic Treatment of Hepatocellular Carcinoma"

_ijms, 2021, doi:10.3390/ijms222313051_

Round 1

Reviewer 1 Report

I read with great interest the manuscript and I found it well written, important in its field and timely. The Authors also introduce some representative images and cartoons that improve the readability and comprehension of the text.

Only minor grammatical and ortographical errors have been found throughout the text.

Author Response

Thanks for the comments by reviewers. We have revised the manuscript accordingly. The added text is marked by yellow color in the revised manuscript.

Reviewer 1:

I read with great interest the manuscript and I found it well written, important in its field and timely. The Authors also introduce some representative images and cartoons that improve the readability and comprehension of the text.

Only minor grammatical and ortographical errors have been found throughout the text.

The authors would like to thank reviewer 1 for their positive feedback and encouragement. The manuscript has now been checked for grammatical and spelling errors.

Reviewer 2 Report

This is a well-compiled article highlighting the role of role of the tumour microenvironment and lymphatic system in the cirrhotic liver, its potential response to TACE, and current and possible novel DDSs for locoregional treatment. Few suggestions for improvements are as follows:

  1. The authors should also clearly highlight the novelty as few articles have already been previously published on similar topic.
  2. The pharmacological strategies to target the tumor microenvironment and lymphatic system in the cirrhotic liver can also be discussed.
  3. The authors should provide their own justification and relevance of the study. This will help the readers to understand the importance of the paper.
  4. Please write the full forms of the abbreviations used in the manuscript and check the manuscript carefully for typos and grammatical errors.

Author Response

Reviewer 2:

This is a well-compiled article highlighting the role of role of the tumour microenvironment and lymphatic system in the cirrhotic liver, its potential response to TACE, and current and possible novel DDSs for locoregional treatment. Few suggestions for improvements are as follows:

The authors would like to thank reviewer 2 for their positive feedback and encouragement.

  1. The authors should also clearly highlight the novelty as few articles have already been previously published on similar topic.

Add the following sentence to the manuscript:

In this review, we have provided a novel and important overview of clinical data, as well as a deeper understanding of the underlying biological mechanisms that could contribute to the response to TACE. In addition, we also highlight the importance of the tumor microenvironment and lymphatic system in this complex interplay.

  1. The pharmacological strategies to target the tumor microenvironment and lymphatic system in the cirrhotic liver can also be discussed.

This following paragraph is added (yellow marked) to the manuscript:

The tumor microenvironment is known to be an active contributor to the progression of HCC. It is therefore not surprising that several treatments have been developed that target the dynamic fibrotic environment that surrounds HCC-nodules. For instance, drugs targeting TGF-β (LY2157299), which is a growth factor highly involved in stellate cell activation, have been tested in clinical trials (NCT01246986 and NCT02178358). Another important contributor to the microenvironment is the deposition of ECM, which includes heparan sulfate proteoglycans. PI-88 is a heparin sulfate mimic that specifically targets heparanase in cancer, thus preventing the release of growth factors that otherwise would contribute to tumor growth, angiogenesis and metastasis (PubMed ID: 10416607). The safety and efficiency of PI-88 as an adjuvant therapy for post-operative HCC has been shown in a phase II trial[ and a recent follow up study revealed significant clinical benefits for patients with HCC (PMID: 1930316), which has been further studied in a phase III trial (NCT01402908). However, none of these compounds have been tested in combination with TACE, yet it could be speculated that reducing the dense network of activated stellate cells and deposition of ECM, would positively influence response to TACE in patients. More studies are needed to confirm this.

Significant progress on the treatment of advanced HCC has been made possible by sorafenib, which in itself targets the tumor microenvironment by blocking angiogenesis. While TACE and sorafenib have both been shown to prolong survival in patients with unresectable hepatocellular carcinoma, previous trials assessing the combination of TACE and sorafenib had been unable to show clinical benefit compared with TACE alone. A recent study from Kudo et al reports the results from a randomized, multi-centre prospective trial of this combination treatment in patients with unresectable HCC. Median progression-free survival was significantly longer in the combination group (25.2 months) than in the TACE without sorafenib group (13.5 months), thus showing promising results in tackling both the tumor cells and the microenvironment in the context of TACE.

  1. The authors should provide their own justification and relevance of the study. This will help the readers to understand the importance of the paper.

At the very end of this manuscript we have added these two sentences:

This report has described the limitations and possibilities of transarterial chemother-apeutic treatment of HCC. It emphasize certain areas to focus on in order to improve TACE treatment for cancer as HCC is expected to increase its incidence in the near future.

  1. Please write the full forms of the abbreviations used in the manuscript and check the manuscript carefully for typos and grammatical errors.

They are already added in a paragraph just prior the reference list.